# A Practical Guide to Source and Receiver Locations for Surface Wave Transmission Measurements across a Surface-Breaking Crack in Plate Structures

**DOI:** 10.3390/s19173793

**Published:** 2019-09-01

**Authors:** Janghwan Kim, Seong-Hoon Kee, Jin-Wook Lee, Ma. Doreen Candelaria

**Affiliations:** 1Department of Civil Engineering, Gwangju University, Gwangju 61743, Korea; 2Department of Architectural Engineering, Dong-A University, 37, Nakdong-Daero 550 Beon-gil Saha-gu, Busan 69315, Korea; 3Advanced Railroad Civil Engineering Division, Korea Railroad Research Institute, 176, Cheoldobangmulgwan-ro, Uiwang-si, Gyeonggi-do 16105, Korea

**Keywords:** surface wave transmission, plate-like structures, surface-breaking crack, air-coupled sensors, multiple bottom-reflected bulk waves

## Abstract

The main objectives of this study are to investigate the interference of multiple bottom reflected waves in the surface wave transmission (SWT) measurements in a plate and to propose a practical guide to source-and-receiver locations to obtain reliable and consistent SWT measurements in a plate. For these purposes, a series of numerical simulations, such as finite element modelling (FEM), are performed to investigate the variation of transmission coefficient of surface waves across a surface-breaking crack in various source-to-receiver configurations in plates. Main variables in this study include the crack depths (0, 10, 20, 30, 40 and 50 mm), plate thicknesses (150, 200, 300, 400 and 800 mm), source-to-crack distances (100, 150, 200, 250 and 300 mm) and receiver-to-crack distances. The validity of numerical simulation results was verified by comparison with results from experiments using Plexiglas specimens using two types of noncontact sensors (laser vibrometer and air-coupled sensor) in the laboratory. Based on simulation and experimental results in this study, practical guidelines for sensor-to-receiver locations are proposed to reduce the effects of the interference of bottom reflected waves on the SWT measurements across a surface-breaking crack in a plate. The findings in this study will help obtain reliable and consistent SWT measurements across a surface-breaking crack in plate-like structures.

## 1. Introduction

Previous researchers have demonstrated that surface wave transmission (SWT) measurements are effective for identifying and characterizing the depth of surface-breaking cracks in solid media [1,2,3,4,5]. The SWT method is based on measurements of surface waves across a surface-breaking crack. Figure 1 shows a typical setup for SWT measurements, consisting of one source and two receivers located on either side of a surface-breaking crack. Incident surface waves generated by the source, *R_i_*, are measured by the near receiver (receiver 1 in Figure 1). When incident surface waves propagate across a surface-breaking crack, the low frequency components of the incident surface waves are transmitted to the forward scattering field with attenuation, while the high-frequency components will be reflected. The transmitted surface waves, *R_tr_*, are measured by the far receiver (receiver 2 in Figure 1). The transmission coefficient of the surface waves is defined as the spectral amplitude ratio between the transmitted surface waves, *R_tr_*, and the incident surface waves, *R_i_*, in the frequency domain (*Tr* = *R_tr_*/*R_i_*). The depth of a surface-breaking crack can be estimated from the pre-established relation between the transmission coefficient, *Tr*, and crack depth, *h*.

Many studies have proposed the relationship between the transmission coefficient of surface waves across a surface-breaking crack, *Tr*, and the normalized crack depth, *h/λ_R_* (crack depth normalized by wavelength of incident surface waves). Achenbach and his colleagues [2,6,7] established the relationship between *Tr* and *h/λ_R_* based on diffraction and scattering of harmonic incident surface waves by a surface-breaking crack in the far-field region of a crack (i.e., the receivers are located far from a crack opening). Cheng and Achenbach [8] demonstrated from experimental studies that *Tr* measured in the far-field of a crack in aluminum specimens (sensor-to-crack distance *x_SC_* ~ 5 *λ_R_*) converges to the analytic solution [2,6,7]. Hirao et al. [9] presented transmission coefficients in a wide range of *h/λ_R_* ($h/\lambda_{R}\in \left[0,\;3\right]$) based on finite element numerical simulations (FEM) and experimental measurements. Masserey and Mazza [10] obtained a transmission function using the finite difference method (FDM) and experimental studies from aluminum specimens. For concrete, a heterogeneous but statistically isotropic material, the SWT method has been proven to be sensitive to the depth of surface-breaking cracks. Hevin et al. [11] obtained the transmission ratio of a surface wave in the frequency domain using boundary element method (BEM), and calculated averaged transmission functions from many different sensor locations. Song et al. [4] obtained the *Tr* and *h/λ_R_* relation based on numerical simulation (BEM) and experimental studies in the laboratory. Recently, Kee and Zhu [12] obtained transmission functions through numerical simulations (FEM) and experimental studies in the laboratory. In the experimental study, air-coupled sensors were used to improve test speed and accuracy in the surface wave transmission measurements.

It has been demonstrated that the location of receivers should be carefully selected to obtain reliable and consistent surface wave transmission measurements. Previous researchers have observed that transmission coefficient *Tr* can be significantly enhanced and very sensitive to the location of sensors when sensors are too close to the crack [12,13,14,15,16]. This near-field scattering effect is mainly caused by the interference of bulk waves (mode converted P- and S waves in front of a surface-breaking crack and bulk waves generated from a tip of a crack) and the direct contribution of incident surface waves. Yew et al. [17] suggested that the location of sensors should be at least the crack depth to minimize the field effect. Chen and Achenbach [8] observed that *Tr* converged to the far-field analytic solution when sensors were located 5$\lambda_{R}$ from a crack opening. Kee and Zhu [13] observed that the near-field effect are reasonably suppressed when the sensor are located 2$\lambda_{R}$ from a crack opening, with error in *Tr* of less than 10%. 

However, most of the studies in the literature were based on the SWT measurements in a solid medium with a half-space assumption, in which the thickness of a plate is assumed to be very large compared to wavelengths of surface waves. The half-space assumption would be valid for the SWT measurements in metallic materials (homogeneous and isotropic materials such as steel and aluminum) since the wavelengths of surface waves (in the order of a few millimeters) are small enough compared to the thickness of the materials. However, for the application to the composite materials (statically isotropic but heterogenous materials such as concrete), the wavelengths of surface waves are generally on the order of a few centimeters, in order to avoid considerable attenuation of signals. In this case, surface waves in a plate could interfere with multiple reflections of P and S waves from the bottom surface of the materials (bottom reflected bulk waves), which affect the transmission coefficient of the surface waves across a surface-breaking crack [18]. One possible way to minimize the contribution of the bottom reflected bulk waves in the transmission calculation is to resolve all possible waves in the time and/or frequency domains. However, this approach requires a large number of sensors [19], which could be an expensive option in the laboratory and the field. As an alternative, a previous study defined and used a transmission function that results from the propagation of all contributing modes for the fixed sensor location [20]. There are many variables that affect SWT coefficients such as thickness of a plate, duration of the impact, material properties of a plate (elastic modulus, Poisson’s ratio and mass density), and source-to-receiver locations. Consequently, the approach based on the fixed sensor location is only valid in special cases with same materials and thickness of a plate. As yet, it is still difficult to find previous research in the literature that proposes a practical guide to source-and-receiver locations to suppress the effect of interference of the bottom reflected bulk waves in the SWT measurements in a plate. 

The primary purposes of this study are to investigate the interference of the multiple bottom reflected bulk waves in the SWT measurements in a plate and to propose a practical guide to source-and-receiver locations to obtain reliable and consistent SWT measurements in a plate. For these purposes, a series of numerical simulations (FEM) is performed to investigate the variation of transmission coefficient of surface waves across a surface-breaking crack in a plate with various source-to-receiver configurations. The main variables in this study include crack depths (0, 10, 20, 30, 40 and 50 mm), plate thicknesses (150, 200, 300, 400 and 800 mm), source-to-crack distances (100, 150, 200, 250 and 300 mm) and receiver-to-crack distances. The validity of FE models was verified by comparison with experiments in Plexiglas specimens using two types of noncontact sensors (laser vibrometer and air-coupled sensor) in the laboratory. Based on the test and simulation results in this study, a practical guideline to source-to-receiver configuration is proposed to reduce the effects of near-field scattering and the interference of bottom reflected bulk waves on the SWT measurements across a surface-breaking crack in a plate. 

## 2. Method

### 2.1. Numerical Simulation Using FEM

The finite element method was used to simulate the transient propagation behavior and near-scattering of surface waves caused by a surface-breaking crack in Plexiglas with a finite thickness. The FE model described in this study was developed according to previous research [12,13,18]. A two-dimensional (2D) FE model was developed using rectangular quadratic plane stress elements (CPS4R) implemented in a finite element package (ABAQUS 2017, Johnston, RI, USA) at Dong-A University, as shown in Figure 2. The main variables in the numerical simulation include the plate thicknesses (100, 150, 200, 300, 400 and 800 mm), the crack depths (0, 10, 20, 30, 40 and 50 mm), source-to-crack distances *x_SC_* (100, 150, 200, 250 and 300 mm) and various receiver locations. In each FE model, two sets of receivers are located on either side of a crack, with various receiver-to-crack distances *x_RC_* ranging from 2 mm to *x_SC_*. Table 1 summarizes the parameters of numerical simulations in this study. In the FE model, two sets of nodes share same coordinate at *x* = 0. Tie constraints were applied at *x* = 0 and h≤y≤H so that compatibility conditions were set as follows,
(1)$$\begin{array}{c} u_{x}\left(0^{-},y\right)=u_{x}\left(0^{+},y\right)\\ u_{y}\left(0^{-},y\right)=u_{y}\left(0^{+},y\right) \end{array}\}\;\;\;\;\;for\;\;\;\;h\leq y\leq H$$
where ***u_x_*** and ***u_y_*** are displacement components of nodes in the *x* and *y* directions, respectively. Consequently, a surface-breaking crack in the FE model was simulated by the discontinuity of two sets of nodes at *x* = 0 and 0≤y≤h (see Detail B in Figure 2).

The mesh size was designed to be 2 mm, so that at least 25 elements can contribute to expressing the minimum wavelength *λ_R_*. In addition, the time increment Δ*t* for integration was determined to be 0.1 μs to maintain stability and accuracy of solution in accordance with Zerwer et al [21]. To reasonably simulate energy dissipation per cycle, material damping was also defined based on linear Rayleigh damping model (***D*** = *η*_1_/2ω+ *η*_2_ω/2, where ***D*** is damping ratio, *η*_1_, and *η*_2_ are constants for mass and stiffness, and ω is angular frequency of wave). The constants *η*_1_ and *η*_2_ were set to 2700 and 5 × 10^−8^ for Plexiglas specimens so that ***D*** was approximately 0.015 in the frequency range of 10 to 50 kHz. 

A transient impact source was applied on the free surface at the location *x_SC_* (see Figure 2). The force function of the transient impact point source is
(2)$$f\left(t\right)=\{\begin{array}{l} \sin^{3}\left(\frac{\pi t}{T}\right)\;\;\;0\leq t\leq T\;\;\;\\ \;\;0\;\;\;\;\;\;\;\;\;\;\;\;\;\;\;\;\;\;\;\;\;\;t>T \end{array}$$
where *T* is the duration of transient force. The cubic force function in Equation (2) was verified to be effective in simulating the transient contact forces by a previous research [22]. The material properties of Plexiglas were assumed to be homogeneous and linearly elastic. This is valid and reasonable within the frequency range covered in this study and reduces the complexity of numerical simulation. The material properties of Plexiglas were selected as follows: Young’s modulus *E* of 5.8 GPa, Poisson’s ratio *v* of 0.33, and mass density *ρ* of 1200 kg/m^3^. The theoretical P and S wave velocities determined by Equations (3) and (4) were 2329 m/s and 1348 m/s, respectively; the corresponding Rayleigh surface wave velocity determined by an approximate expression in Equation (5) was 1256 m/s [23].
(3)$$C_{P}=\sqrt{\frac{E}{\rho \left(1-\nu^{2}\right)}}$$
(4)$$C_{S}=\sqrt{\frac{E}{2\rho \left(1+\nu \right)}}$$
(5)$$C_{R}=\frac{\left(0.87+1.12\nu \right)}{\left(1+\nu \right)}C_{S}$$

In this study, the transmission coefficient of surface waves (acceleration components) across a surface-breaking crack is defined as the spectral amplitude ratio between the incident surface waves measured by far-receiver located at *x_RC_* and transmitted surface waves measured by near-receiver located at −*x_RC_* (see Figure 2). *Tr*, which is measured in various source and receiver configuration, crack depths in FE model with different thicknesses, is expressed as follows,
(6)$$Tr\left(x_{S},x_{R},f,h,H\right)=\frac{S\left(-x_{SC},x_{RC},f,h,H\right)}{S\left(-x_{SC},-x_{RC},f,h,H\right)}$$
where $x_{S}$ is the location of an impact source, $x_{R}$ is the location of receivers, and ***S*** is the Fourier transform of the vertical component of surface waves. To eliminate geometric attenuation effects, the transmission coefficient is normalized by the value obtained from a crack-free model ***Tr*** ($x_{s},x_{R},f,$ 0, *H*). 

Validation of the FE model in this study was verified by a comparison with results measured by a laser vibrometer. The specifics of the comparison of FE results and experiments is described in Appendix A. 

### 2.2. Experiments: SWT Measurements in Plexiglas Specimens

#### 2.2.1. Test Specimen

Plexiglas (poly (methyl methacrylate), PMMA) specimens with a length of 1220 mm, a width of 6.4 mm and a thickness of 200 mm were used in this study. The width of the specimens was less than 1/10 of the shortest wavelength of propagating waves (*λ_min_*/*C_p_* ~ 0.09 in a frequency range from 5 to 35 kHz). Therefore, the plain stress condition could be applied for the propagation of stress waves [5]. The specimens were set in upright position as shown in Figure 3. A notch-type crack was created using a hand saw in the Plexiglas specimen, with the crack depth *h* increasing from 0 to 40 mm, in increments of 10 mm. The hand saw made approximately 0.5-mm-wide cracks, resulting in a width-to-depth ratio smaller than 0.1 for all cracks in this study. This value is small enough to ignore the effect of the crack width on the surface wave measurements across the crack so that the experimental results could be directly compared with the theoretical analysis results [10]. 

#### 2.2.2. Test Setup, Data Acquisition and Signal Processing for the SWT Measurements

Two air-coupled sensors (PCB model No. 377B01, Depew, NY, USA) at the University of Texas at Austin were used to investigate the variation of surface wave transmission coefficient across a surface-breaking crack in the Plexiglas specimen with varying source-to-receiver distances, *x_SR_*, and receiver-to-crack distances, *x_RC_*. Steel balls with two different diameters (5 mm and 10 mm) were used to generate impact-induced surface waves with different frequency contents, with center frequencies of 25 and 15 kHz, respectively. 

Transmission coefficients of the surface waves were measured based on a self-calibrating procedure proposed by Achenbach et al. [24] and Popovics et al. [3]. To investigate the effects of receiver locations of transmission measurements of surface waves, the distance of each receiver from a crack opening, *x_RC_*, varied from ±10 to ±150 mm with an increment of ±10 mm (see Figure 3). Figure 4a shows typical time signals obtained from crack-free Plexiglas specimens using air-coupled sensors located at *x_RC_* = ±70 mm by applying an impact at *x*_S_ = −250 mm, where *x*_S_ is the location of an impact source. Original signals and windowed signals are indicated with dash lines and solid lines, respectively. Windowed signals were obtained by applying Hanning window with the size of 4 times of the period at the minima of surface wave components. Transmission coefficients were calculated in the frequency domain using Equation (7)
(7)$$Tr\left(x_{S},x_{R},f,h,H\right)=\sqrt{\frac{S\left(-x_{SC},x_{RC},f,h,H\right)S\left(x_{SC},-x_{RC},f,h,H\right)}{S\left(-x_{SC},-x_{RC},f,h,H\right)S\left(x_{SC},x_{RC},f,h,H\right)}}$$
where ***Tr*** (*x_s_*, *x*_R_, f,
*h*, *H*) is the transmission coefficient of surface waves across a surface-breaking crack with a depth of *h* in a plate with a thickness of *H*. The surface waves were generated by an impact source made by authors located at *x_s_* and measured by two receivers (air-coupled sensors) located at the either side of the crack (±*x_RC_*) and ***S***(*x_s_*, *x*_R_
,f,
*h*, *H*) is the Fourier transform of the windowed time signal generated by an impact source at *x_s_* and measured by the sensor at *x*_R_. Figure 4b shows the ***Tr*** versus frequency obtained from Plexiglas specimens having a surface breaking crack with depths of 0, 10, and 20 mm. As the depth of a crack increases, high frequency component of surface wave decreases. 

Song et al. (2003) and Kee and Zhu (2010) demonstrated that the self-calibrating procedure was effective for eliminating experimental variability caused by impact sources and receivers. They also showed that the effects of geometric attenuation and material damping could be mitigated by normalizing ***Tr**_h_* with a transmission coefficient obtained from crack-free specimens (***Tr***_0_).
(8)$$Tr_{n}=Tr_{h}/Tr_{0}$$

In addition, to improve signal-to-noise level, spectral coherence curves were also calculated based on five repeated signal data collected at the same location using Equation (9).
(9)$$SC_{12}=\frac{{\left|\sum G_{12}\right|}^{2}}{\sum G_{11}\sum G_{22}}$$
where ***G***_12_, ***G***_11_ and ***G***_22_ are the cross spectrum and auto spectrum functions between the time signal ***S***(*x_s_*, −*x*_R_
,f,
*h*, *H*) and ***S***(*x_s_*, *x*_R_
,f,
*h*, *H*) that are measured by receiver 1 and receiver 2, respectively. Similarly, ***SC_21_*** can also be calculated from signals ***S***(*x_s_*, *x*_R_
,f,
*h*, *H*) and ***S***(*x_s_*, −*x*_R_
,f,
*h*, *H*). The value of ***SC*** ranges from 0 to 1.0, where a value close to 1.0 indicates good signal quality and repeatability. Acceptable frequency ranges, where spectral coherence is higher than 0.99, were determined. Figure 4b shows typical SC function obtained from Plexiglas specimens having a surface breaking crack with depths of 0, 10, and 20 mm by an impact of a 5 mm diameter steel ball. According to the signal coherence function in this study, the acceptable frequency range was determined to be in the range of 15 to 35 kHz and 10 to 20 kHz for the signals generated by 5-mm- and 10-mm-diameter steel balls, respectively.

## 3. Result and Discussion

### 3.1. Interaction of Surface Waves and Multiple Bottom Reflected Waves

Figure 5a,b shows B-scan images representing vertical velocity responses of impact-induced elastic stress waves obtained from a crack-free FE model and a Plexiglas specimen in the laboratory. The thickness of plates in the FE simulation and experiment was 200 mm. The B-scan images were constructed by stacking normalized time-domain vertical responses, ***V*_y_**(*x*_R_, *t*)/max|***V*_y_**(*x*_R_, *t*)|, obtained from various receiver locations, with *x*_R_ ranging from 100 mm to 360 mm. Time signals used in Figure 5b were measured by a laser vibrometer in the laboratory. Specifics on the test setup and test procedure using a laser vibrometer are described in Appendix A. For comparison purposes, first arrivals of possible waves were determined using the seismic reflection theory [25,26] and presented as dashed lines in Figure 5a,b. Direct P, S, and surface waves are denoted as *P_i_*, *S_i_*, and *R_i_*, and several multiple reflected bulk waves are denoted as *PP_n_*, *SS_n_*, *PS_n_*, *PPPS_n_*, and *SSSP_n_*. Overall, the B-scan image based on the normalized velocity responses from the FE model shows good agreement with the experimental measurements. In particular, the first arrivals of the propagating waves in a plate match well with those determined by the seismic reflection theory. It was observed from the B-scan images that the surface waves could interfere with multiple bottom reflected bulk waves. Observation shows that the interference of multiple reflected bulk waves in the surface wave measurements depends on several influencing parameters such as thickness of a plate *H*, source-to-receiver distance *x_SR_* and duration of impact source *T*.

In this study, the amplification of surface waves was used to more precisely investigate the interference of surface waves with the bottom reflected bulk waves in a free plate. The amplification coefficient of surface waves, *α*, is defined as the relative maximum amplitude of absolute acceleration components of the surface waves in a plate with a thickness of *H* compared to that in a plate with a thickness of 800 mm as follows,
(10)$$\alpha =\frac{\max\left|A_{y,win}\left(x_{S},x_{R},t,h,H\right)\right|}{\max\left|A_{y,win}\left(x_{S},x_{R},t,h,800\right)\right|}$$
where $A_{y,win}\left(x_{S},x_{R},t,h,H\right)$ is vertical acceleration components of surface waves, generated by an impact source at $x_{S}$ and measured by a receiver at $x_{R}$, across a surface-breaking crack with a depth of *h* in a plate with a thickness of *H*. The surface wave component was extracted from a full waveform by applying window function as described in the previous section. Please note that the plate with a thickness of 800 mm represents the half-space model in this study, since the first arrivals of PP waves are completely separated from the surface waves in *x_SR_* ranging from 40 to 360 mm.

Figure 6a,b shows the variation of amplification coefficient *α* obtained from the crack-free FE models (*h* = 0 mm) and the cracked models (*h* = 30 mm) in Table 1, respectively, with respect to normalized source-to-receiver location, *x_SR_*/*H* (source-to-receiver distance normalized by the thickness of a plate). It was observed that the effect of bottom reflected waves was significant when *x_SR_* is greater than about the thickness of a plate, *H*. 

Theoretically, surface waves can be separated from the first bottom reflection wave PP_1_ in the time domain in the following condition
(11)$$t_{r}+T\leq t_{PP1},$$
where $t_{r}=x_{SR}/C_{R}$, $t_{pp1}=2\sqrt{H^{2}+{\left(x_{SR}/2\right)}^{2}}/C_{p}$ and *T* is duration of surface waves. Inserting Equations (3) and (4) into Equation (11) and rearranging the equation results in
(12)$$\left(\eta^{2}-1\right){\left(\frac{x_{SR}}{H}\right)}^{2}+\frac{2\eta^{2}\lambda_{R}}{H}\left(\frac{x_{SR}}{H}\right)+{\left(\frac{\eta \lambda_{R}}{H}\right)}^{2}-4\leq 0\;\mathrm{and}\;\frac{x_{SR}}{H}\geq 0$$
where *η* = $\frac{C_{P}}{C_{R}}$ and $\lambda_{R}$ is wavelength of surface waves ($\frac{T}{C_{R}}$). Therefore, it can be said that surface waves propagating in a plate can be separated from the first bottom reflected bulk wave PP1 when source-to-receiver distance satisfies the following condition,
(13)$$0\leq \frac{x_{SR}}{H}\leq \beta$$
where *β* is a constant dependent on the thickness of a plate, *H*, wavelength of surface wave, $\lambda_{R}$ and Poisson’s ratio of material, *v*. Figure 7a shows that the variation of minimum value of β with varying *H* and *v* for *T* = 30 μs. It was observed that interference of surface waves and bottom reflected waves can be reasonably avoided when $x_{SR}\sim H$ if the thickness of a plate is greater than a critical value, *H_cr_*. Figure 7b shows the variation of *H_cr_* with *v* in the range of 0.15 to 0.4 and *T* in a range of 30 μs to 60 μs. It was observed that normalized critical thickness, *H_cr_*/$\lambda_{R}$ (critical thickness normalized by wavelength of surface wave) is dependent on Poisson’s ratio *v*. An approximate formula relating *H_cr_*/$\lambda_{R}$ and *v* was established by non-linear regression analysis as follows,
(14)$$H_{cr}/\lambda_{R}=47.5\;\nu^{2}-14.1\nu +4.5.$$

As a rule of thumb, the source and receiver distance to avoid the interference of multiple bottom reflected bulk waves is recommended as follows,
(15)$$x_{SR}\leq H\;\mathrm{and}\;H\geq H_{cr}$$
For the Plexiglas specimen, $H_{cr}/\lambda_{R}$ is about 5.0, which means that the simplified rule is only valid when the thickness of a plate is greater than about $5\lambda_{R}$.

### 3.2. Effect of Bottom Reflected Waves on Tr_n_

Figure 8a,b shows the variation of the normalized transmission coefficient of surface waves with two relative sensor locations (normalized source-to-receiver distance, *x_SR_/H* and normalized receiver-to-crack distance, *x_RC_/λ_R_*). The transmission coefficients calculated at the center frequencies of Plexigals specimens, 15 and 25 kHz, are depicted as open circles in Figure 8a,b, respectively. For comparison purposes, the transmission coefficient from the FE models are also shown in the same figures as lines. Overall transmission coefficients measured by experiments shows good agreement with numerical simulation results. Both experimental and numerical simulation results reveal that the surface wave transmission coefficients depend on the normalized crack depth (*h/λ_R,_* the depth of a surface-breaking crack normalized by the wavelength of surface waves) and the two relative sensor locations, *x_SR_/H* and *x_RC_/λ_R_*.

Consistent with observations from previous researches, the transmission coefficient of surface waves is enhanced compared to those measured in the far-field [13,15,16] when sensors are located close to a surface-breaking crack. This is a consequence of the interaction of surface waves with a surface-breaking crack, which is called the near-field scattering of surface waves. Previous researchers have demonstrated that the near-field effect in a half-space model is reasonably suppressed when the sensors are located farther than about two times of the wavelength of surface wave (*x_RC_* ≥ 2*λ_R_*) [13]. However, *Tr_n_* values in a plate with finite thicknesses show oscillatory behavior with considerable amplitude, even when the receivers were farther distant than 2*λ_R_*. This phenomenon indicates that the criteria of receivers to avoid the near-field effect in the half space model may not work in the SWT measurements across surface-breaking crack in a plate.

Figure 9 shows that the variation of transmission coefficient in free plates with various thicknesses (800, 400, 300, 200, and 150 mm). The thickness of a plate affects the transmission coefficient of surface waves across a surface-breaking crack. As the thickness of a plate decreases, the transmission coefficient of surface waves tends to deviate from those from the thick plate. Figure 10 shows the difference of ***Tr****_n_* obtained from FE models with various thicknesses, *H* = 150, 200, 300, and 400 mm, (=|***Tr****_n_*(*H*) − ***Tr****_n_*(*H* = 800mm)|) and various source-to-receiver locations in the range of 0.1 to 3.6 compared to ***Tr***_*n*_ from a plate with a thickness of 800 mm (approximate half-space model in this study). The results from FE models with three different source locations (*x_SC_* = 100, 200, and 250 mm) are presented as black, blue and red open circles. The effect of bottom reflected waves on the transmission coefficient is reasonably suppressed when the source-to-receiver distance of a far sensor (receiver 2) is less than *H*. This result demonstrated the validity of the simplified rule for the source-to-receiver location for avoiding the effect of bottom reflection in SWT measurements (see Equation (15)). 

### 3.3. Effect of Bottom Reflected Waves on Crack Depth Estimation 

Figure 11 shows the normalized transmission coefficient *Tr_n_* and *h/λ_R_* relationship measured from a FE model with a thickness of 800 mm. The results were obtained from the sensors located at least 4 *λ_R_* from a crack opening, and far sensors located within 0.5*H* from the source location. Consequently, the *Tr_n_* and *h/λ_R_* relation in Figure 11 can be regarded as a far-field surface wave transmission measurement across a surface-breaking crack in a half-space Plexiglas model. The *Tr_n_* versus *h/λ_R_* curve from the FE model in this study was approximated by a polynomial function using a non-linear regression as follows,
(16)$$tr_{n}=\overset{6}{\underset{i=0}{\sum }}c_{i}(\frac{h}{\lambda R}{)}^{i}$$
where *c*_1_ to *c*_6_ are constants (*c*_0_ = 1, *c*_1_ = 0.9213, *c*_2_ = −19.75, *c*_3_ = 54.68, *c*_4_ = −65.99, *c*_5_ = 37.51, and *c*_6_ = −8.215) determined by the curve fitting of the numerical results. 

In the surface wave transmission measurements, the approximate expression that describes the *Tr_n_* and *h/λ_R_* can be used to predict the depth of a surface-breaking crack. By using a broadband impact source, many transmission values are obtained within the frequency range of interest. Thus, multiple redundant estimates of the crack depth may be calculated from a single measurement. In this study, the most likely depth of a surface-breaking crack was determined by using the least square method. The estimated depth was determined by minimizing the root mean square error (RMSE),
(17)RMSE=∑i=1n[trn(fi,hλRi)−Trn(fi,hλRi)]2
where *tr_n_* is the transmission ratio in the proposed calibration curve in Equation (16), *Tr_n_* is the measured transmission ratio calculated using Equations (7) and (8), *i* is an index of the input values, and *f_i_* and *λ_R__i_* are the frequency and wavelength for the *i*-th component.

Figure 12 is a 2D scattering plot representing the error in predicted crack depth, *Error_depth_*, using SWT measurements across a surface-breaking cracks (*h* = 0, 10, 20, 30, 40, and 50 mm) in a plate with various thicknesses (*H* = 100, 150, 200, 300, 400, and 800 mm) with normalized source-to-receiver (far receiver or receiver 2) location, *x_SR,F_*/*H*, and normalized sensor-to-crack distance, *x_RC_*/*λ_R_*. In this study, *Error_depth_* was defined as difference between predicted crack depth, $h_{pred}$, compared to the true depth, $h_{true}$, as follows,
(18)$$Error_{depth}=\frac{\left|h_{pred}-h_{true}\right|}{h_{true}}\times 100\;\left(\%\right)$$
Figure 12 shows that there are considerable errors between the predicted and the true crack depths, *Error_depth_*, when receivers are located too close to the sensor which is mainly due to the near-field effect of receivers. In addition, *Error_depth_* could be noticeable when receivers are located too far from the source compared to the thickness of a plate which is consequences of interaction of surface wave and bottom reflected bulk waves in a plate. It was observed that *Error_depth_* was significantly suppressed when *x_RC_*/*λ_R_* was at least 2.0 and *x_SR,F_*/*H* was less than 1.0. 

## 4. Summary and Conclusions

The effect of bottom reflected bulk waves on the surface wave measurements in a plate with various thicknesses was investigated by a series of finite element (FE) simulations. The validity of the FE model was verified with experimental results from a Plexiglas specimen in the laboratory. The specific conclusions obtained from this study are summarized as follows: Consistent with previous research based on the half-space assumption, the transmission coefficient of surface waves in a plate could be significantly enhanced, and the values are sensitive to the location of receivers from crack-opening, which is known as the near-field scattering effect in the SWT measurements. It has been demonstrated in the literature that the near-field effect could be reasonably suppressed when the receivers are located far enough from crack opening (e.g., $x_{RC}\geq$ 2*λ_R_*). However, results based on experiments and numerical simulations in this study exhibit that the approximate far-field criterion based on the half-space assumption is only valid when the effect of bottom reflected bulk waves is not significant. It is recommended in this study that a receiver should be placed not farther than the thickness of a plate from an impact source $\left(x_{SR}\leq H\right)$ to reasonably suppress the interference of surface waves and bottom reflected bulk waves in a plate. Please note that this simplified rule; however, may not be effective for the application to relatively thin plate with a thickness less than a critical thickness, *H_cr_*, which is dependent on Poisson’s ratio and the wavelength of surface waves *λ_R_*. An approximate equation relating the normalized critical thickness, *H_cr_*/*λ_R_* (critical thickness normalized by wavelength of surface waves) and Poisson’s ratio of materials, *v*, was established in this study.It was verified from a series of FE simulations that the practical guideline to source-to-receiver locations and receiver-to-crack locations are effective to obtain reliable and consistent crack depth estimation by the surface wave transmission measurements in a plate.

## Figures and Tables

**Figure 1 sensors-19-03793-f001:**
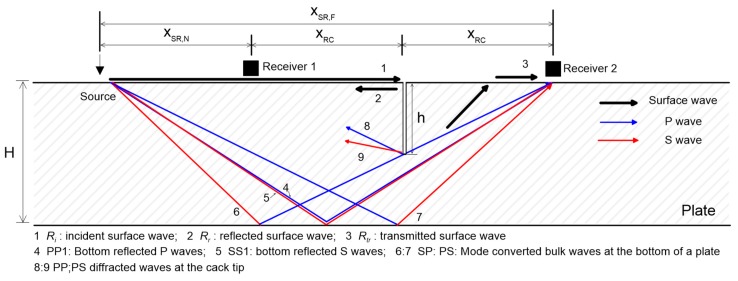
A typical source-to-receiver configuration for surface wave transmission measurements across a surface-breaking crack in a plate.

**Figure 2 sensors-19-03793-f002:**
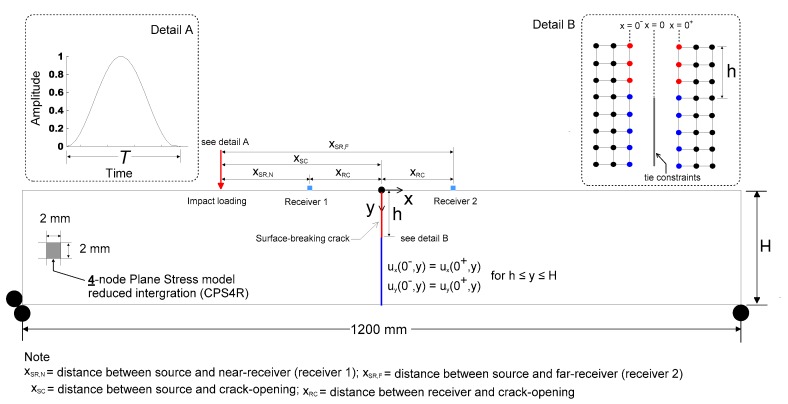
Finite element model of simulating the interaction of surface waves with a surface-breaking crack in a plate.

**Figure 3 sensors-19-03793-f003:**
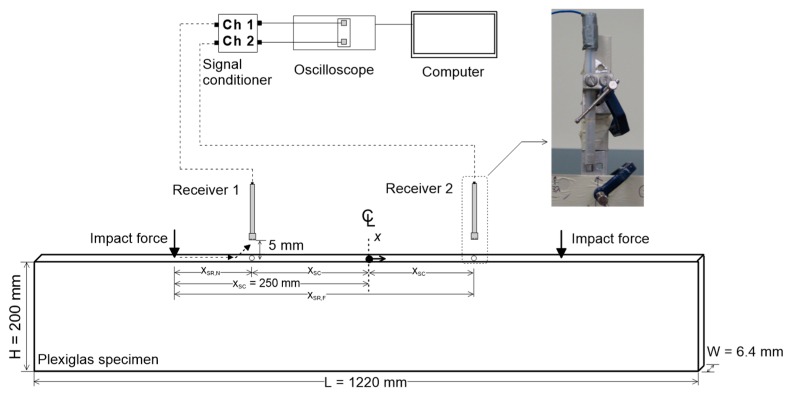
Test setup of air-coupled surface wave measurement in a Plexiglas specimen.

**Figure 4 sensors-19-03793-f004:**
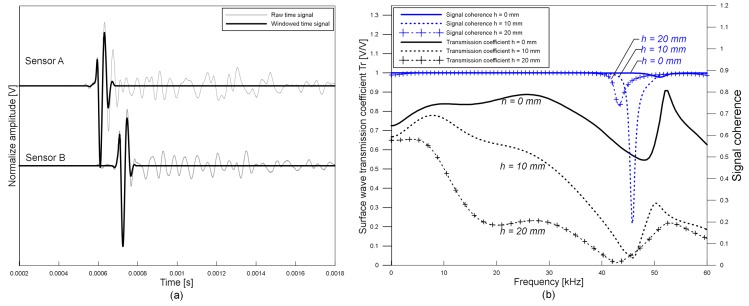
Typical time and frequency signals: (**a**) raw and windowed signals in the time domain, and (**b**) surface wave transmission coefficient and signal coherence in the frequency domain.

**Figure 5 sensors-19-03793-f005:**
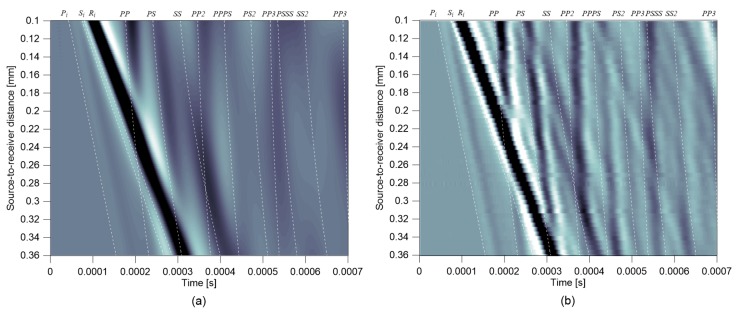
B-scan images representing vertical velocity responses of impact-induced elastic stress waves: (**a**) results from a crack-free FE numerical simulation, and (**b**) results from a Plexiglas specimen measured by a laser vibrometer. The thickness of plates in the FE simulation and experiment in the laboratory was 200 mm.

**Figure 6 sensors-19-03793-f006:**
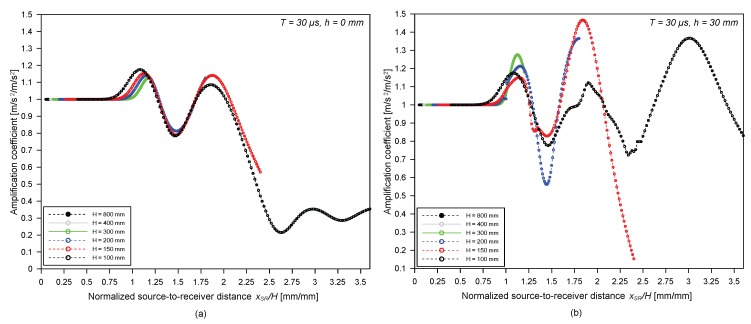
Variation of amplification coefficient of vertical acceleration components of surface waves, *α,* with respect to normalized source-to-receiver distance, *x_SR_*/*H*: (**a**) from the crack-free FE models, and (**b**) the cracked FE models (*h* = 30 mm) with various plate thicknesses—100, 150, 200, 300, 400, and 800 mm.

**Figure 7 sensors-19-03793-f007:**
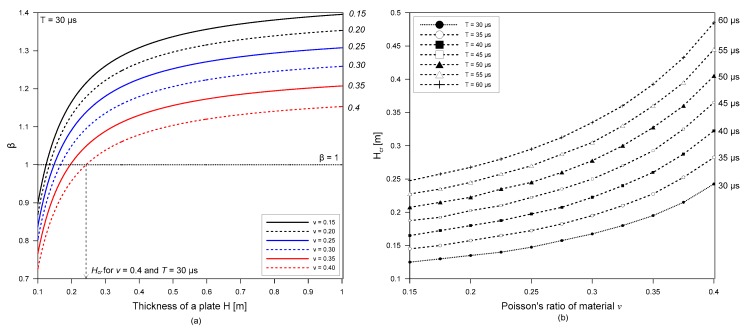
(**a**) Variation of minimum value of ß in Equation (13) with respect to the thickness of a plate, *H*, and Poisson’s ratio, *v* and (**b**) variation of critical thickness, *H_cr_*, with respect to Poisson’s ratio, *v*, and duration of incident waves, *T*.

**Figure 8 sensors-19-03793-f008:**
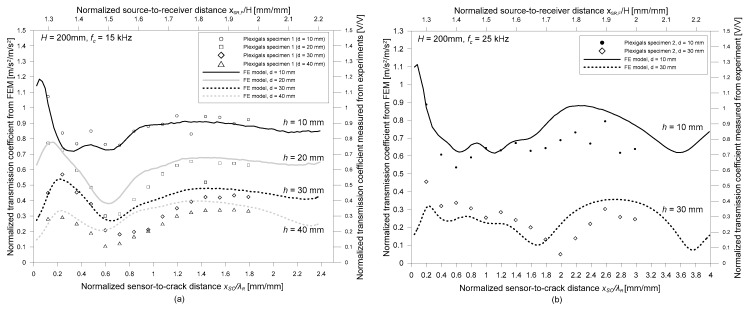
Variation of normalized transmission coefficient of surface waves, *Tr_n_*, in plate with the thickness of 200 mm with respect to two relative sensor locations (normalized source-to-receiver distance, *x_SR_/H* and normalized receiver-to-crack distance, *x_RC_/λ_R_*) at two frequencies: (**a**) *f_c_* = 15 kHz and (**b**) *f_c_* = 25 kHz. For comparison, experimental results measured on Plexiglas specimens with various crack depths are shown in the figure.

**Figure 9 sensors-19-03793-f009:**
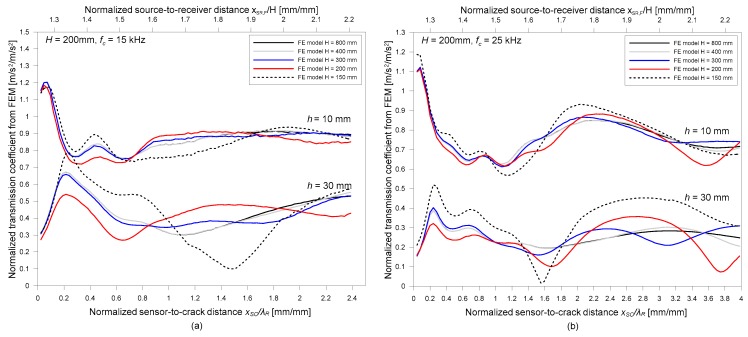
Variation of normalized transmission coefficient of surface waves, *Tr_n_*, in the FE models with various plate thicknesses (*H* = 150, 200, 300, 400, and 800 mm), with respect to two relative sensor locations (normalized source-to-receiver distance, *x_SR_/H* and normalized receiver-to-crack distance, *x_RC_/λ_R_*) at the two frequencies: (**a**) *f_c_* = 15 kHz, and (**b**) *f_c_* = 25 kHz.

**Figure 10 sensors-19-03793-f010:**
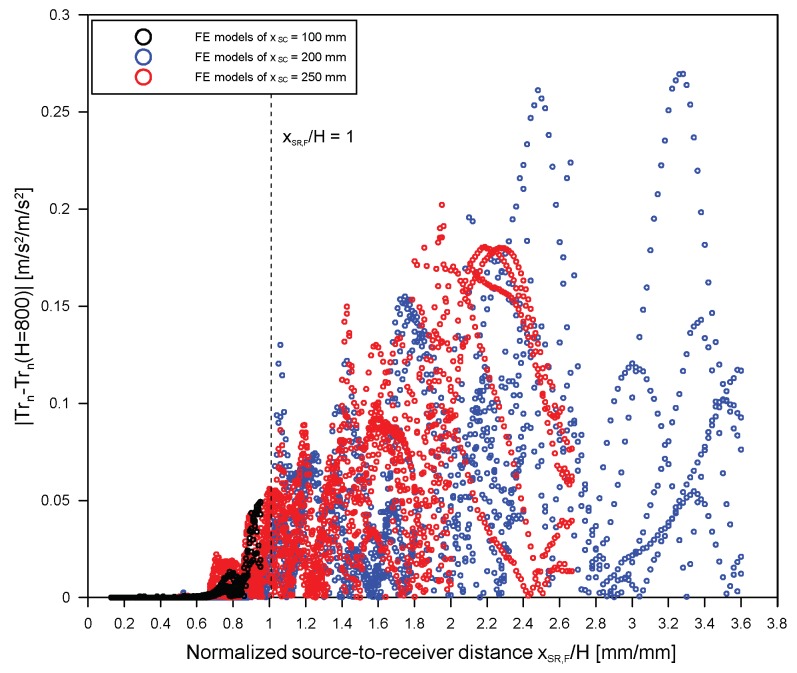
Difference of ***Tr****_n_* obtained from FE models with various thicknesses (*H* = 150, 200, 300, and 400 mm) and various source-to-receiver locations in a range of 0.1 to 3.6 compared to ***Tr****_n_* from a plate with a thickness of 800 mm (approximate half-space model in this study). The results from FE models with three different source locations (*x_SC_* = 100, 200, and 250 mm) are presented as black, blue, and red open circles.

**Figure 11 sensors-19-03793-f011:**
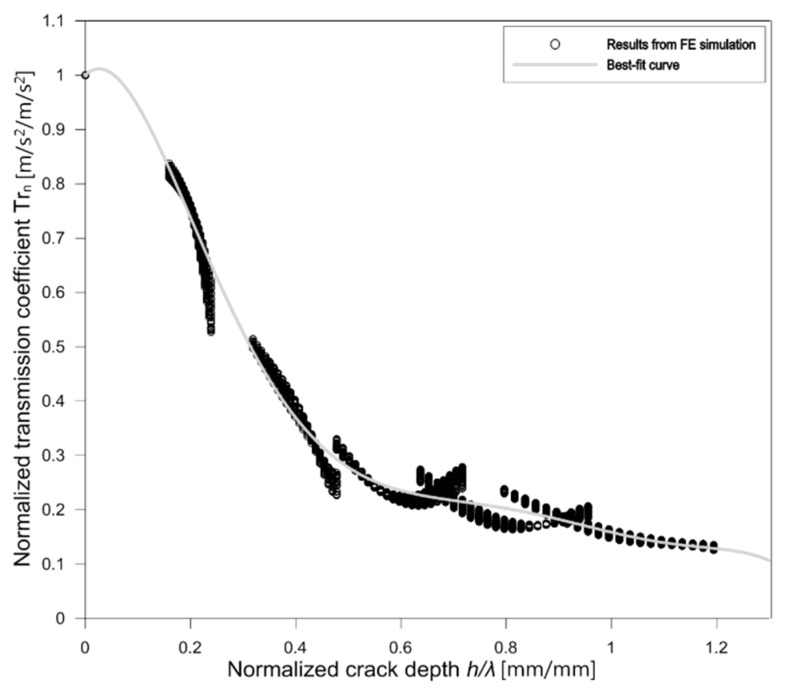
Normalized transmission coefficient of surface waves across a surface-breaking crack in a plate with a thickness of 800 mm (approximate half-space model in this study) versus normalized crack depth.

**Figure 12 sensors-19-03793-f012:**
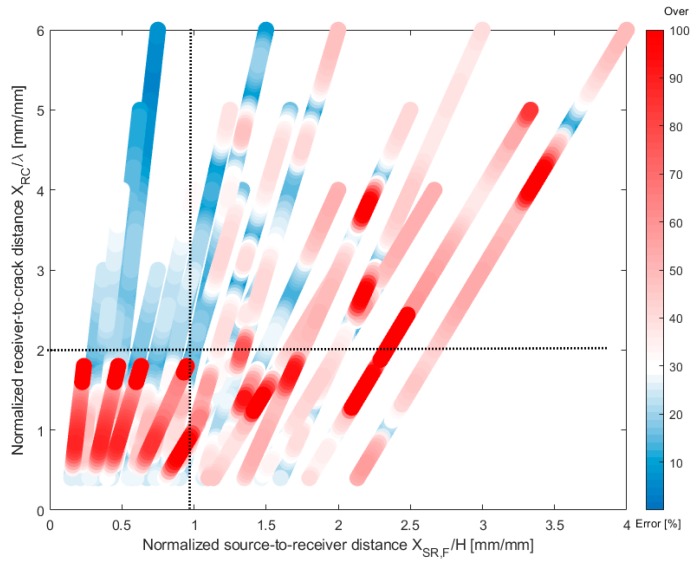
2D scattering plot representing the error in predicted crack depth, *Error_depth_*, using SWT measurements across a surface-breaking cracks (*h* = 0, 10, 20, 30, 40, and 50 mm) in a plate with various thicknesses (100, 150, 200, 300, 400, and 800 mm) with the two normalized receiver locations, *x_SR,F_*/*H*, and *x_RC_*/*λ_R_*.

**Table 1 sensors-19-03793-t001:** Finite element models and parameters.

*v*	*E* (GPa)	*ρ* (kg/m^3^)	*H* (mm)	*H* (mm)	*x_sc_* (mm)	*T* (μs)
0.33	5.8	1200	100, 150, 200, 300, 400, 800	0, 10, 20, 30, 40, 50	100, 150,200, 250, 300	30

Note: *v* = Poisson’s ratio, *E* = modulus of elasticity, *ρ* = density, *H* = thickness of a plate, *h* = depth of a surface-breaking crack, *x_SC_* = source-to-crack distance (see Figure 2), *T* = duration of incident waves.

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
