# Peer review of "A Practical Guide to Source and Receiver Locations for Surface Wave Transmission Measurements across a Surface-Breaking Crack in Plate Structures"

_sensors, 2019, doi:10.3390/s19173793_

Round 1

Reviewer 1 Report

The entire article needs to be reviewed for some stylistic/spelling/grammatical errors. For instance:

1. page 6,row 61/62: "Original signals and windowed signals are indicated with dash lines and solid lines, respectively" - Figure 4 a) legend is reversed with respect to the above mentioned statement.

2. Figure 4 b) legend is slightly shifted upwards without being aligned with the upper bound of the graph.

3. page 7, row 230 "suface" 

4. page 10, row 328 ''surafce''

5. page 10,row 337 and page 11,row353  ''thciknesses''

6. page 12, row 383 - "i component" - I would recommnend "i-th component"

Almost all graphs DO NOT contain units! Please add the units to both axes, even if the variables are dimensionless.

Note: I would recommend to more specify the type of waves in case of, for example, "bottom reflected waves". We generally now, that these waves belong to the bulk waves, but in this case it would be favorable to add the "bulk" term.

Note: page 3, section 2.1. - A surface-breaking crack in the FE model was simulated by discontinuity of two sets of nodes that share same coordinates, but are disconnected (see detail B in Figure 2). - It would be nice to use there a mathematical formulation of the boundary condition instead of/and stated expression.

Note: page 5: The width of the specimens was less than 1/10 of the shortest wavelength of propagating waves (λmin/Vp ~ 0.09 in a frequency range from 5 kHz to 35 kHz). Therefore, the plain stress condition could be applied for the propagation of stress waves. - I would welcome a reference to literature.

Author Response

The authors deeply appreciate a careful review of the reviewer, and completely agree with all the comments of the reviewer. The manuscript was revised according to the reviewer’s comments. Responses to individual reviewer's comments are prepared in a separated file. In addition, a major revision was reflected in the revised manuscript.    

Reviewer 2 Report

The authors present a numerical and experimental study in view to propose optimal sensors placement for surface wave transmission measurement and breathing crack characterization. The paper is generally well written and the paper fit in the scope of the journal.

However, the critical point concerns the novelty of the present work. In my opinion, most of the conclusions were already available in the literature, and especially in the authors' own previous works. In addtion, the conception of the study, the numerical model, the experimental setup, most of the figures are extremely similar (if not identical) to those of ref [12, 13, 18]. For this reason, this paper is in my view not suitable for publication unless the authors can include a clear added-value and explicitly identify the contribution.

Others comments:

Figure 10 should be improved for more clarity (maybe a line at X/H=1 to show the limit where the effect of bottom reflected wave vanishes). Typos: introduction, page 3, line 94 "in special" line 97, "for these" Section 2.1, line 110 "method was"

Author Response

The authors deeply appreciate a careful review of the reviewer, and completely agree with all the comments of the reviewer. The manuscript was revised according to the reviewer’s comments. In particular, the authors carefully reviewed the previous works and clearly identified new contribution of the present work in this study. Responses to individual reviewer's comments are prepared in a separated file. In addition, a major revision was reflected in the revised manuscript.   

Round 2

Reviewer 2 Report

The authors have addressed the comments and I suggest that the paper can now be published in the present form.